# Epidemiological and Genetic Characteristics of Respiratory Viral Coinfections with Different Variants of Severe Acute Respiratory Syndrome Coronavirus 2 (SARS-CoV-2)

**DOI:** 10.3390/v16060958

**Published:** 2024-06-13

**Authors:** Ivelina Trifonova, Neli Korsun, Iveta Madzharova, Ivailo Alexiev, Ivan Ivanov, Viktoria Levterova, Lyubomira Grigorova, Ivan Stoikov, Dean Donchev, Iva Christova

**Affiliations:** 1Department of Virology, National Centre of Infectious and Parasitic Diseases, 1233 Sofia, Bulgaria; neli_korsun@abv.bg (N.K.); iveta.madzharova@abv.bg (I.M.); ivoalexiev@yahoo.com (I.A.); lyubomiragrigorova@gmail.com (L.G.); iva_christova@yahoo.com (I.C.); 2Department of Microbiology, National Centre of Infectious and Parasitic Diseases, 1504 Sofia, Bulgaria; ivanoov@gmail.com (I.I.); vikis@abv.bg (V.L.); ivanstoikovbt@gmail.com (I.S.); deyandonchev@ncipd.org (D.D.)

**Keywords:** respiratory viruses, coinfections, multiplex NGS, SARS-CoV-2, RSV-B, influenza viruses, PIV3, BoV, AdV-C

## Abstract

This study aimed to determine the incidence and etiological, seasonal, and genetic characteristics of respiratory viral coinfections involving severe acute respiratory syndrome coronavirus 2 (SARS-CoV-2). Between October 2020 and January 2024, nasopharyngeal samples were collected from 2277 SARS-CoV-2-positive patients. Two multiplex approaches were used to detect and sequence SARS-CoV-2, influenza A/B viruses, and other seasonal respiratory viruses: multiplex real-time polymerase chain reaction (PCR) and multiplex next-generation sequencing. Coinfections of SARS-CoV-2 with other respiratory viruses were detected in 164 (7.2%) patients. The most common co-infecting virus was respiratory syncytial virus (RSV) (38 cases, 1.7%), followed by bocavirus (BoV) (1.2%) and rhinovirus (RV) (1.1%). Patients ≤ 16 years of age had the highest rate (15%) of mixed infections. Whole-genome sequencing produced 19 complete genomes of seasonal respiratory viral co-pathogens, which were subjected to phylogenetic and amino acid analyses. The detected influenza viruses were classified into the genetic groups 6B.1A.5a.2a and 6B.1A.5a.2a.1 for A(H1N1)pdm09, 3C.2a1b.2a.2a.1 and 3C.2a.2b for A(H3N2), and V1A.3a.2 for the B/Victoria lineage. The RSV-B sequences belonged to the genetic group GB5.0.5a, with HAdV-C belonging to type 1, BoV to genotype VP1, and PIV3 to lineage 1a(i). Multiple amino acid substitutions were identified, including at the antibody-binding sites. This study provides insights into respiratory viral coinfections involving SARS-CoV-2 and reinforces the importance of genetic characterization of co-pathogens in the development of therapeutic and preventive strategies.

## 1. Introduction

Respiratory viruses are a leading cause of morbidity and contribute to persistent public health problems. Influenza A/B viruses, respiratory syncytial viruses (RSVs), rhinovi-ruses (RVs), parainfluenza viruses (PIVs), human metapneumoviruses (HMPVs), ade-noviruses (AdVs), and bocaviruses (BoVs), endemic coronaviruses 229E, OC43, NL63, and HKU-1 have been reported as common causes of seasonal respiratory diseases [1]. Since its emergence in 2019, severe acute respiratory syndrome coronavirus 2 (SARS-CoV-2) has led to a global pandemic [2,3]. During the early stages of the pandemic, there was a simultaneous spread of influenza and SARS-CoV-2 [4] (Maltezou HC, 2023). As the pandemic continued, the highly contagious Delta and Omicron variants of SARS-CoV-2 emerged, while reports of influenza infections remained low [5]. The detection of other non-influenza respiratory viruses was also minimal during this time [6]. After the removal of non-pharmaceutical interventions (NPI) in 2022, there was a slight increase in the occurrence of respiratory infections caused by influenza and other respiratory viruses [7]. By 2023, the prevalence of respiratory viruses had returned to pre-pandemic levels. In the 2023–2024 season, the evidence of SARS-CoV-2 was significantly lower than the previous year [8]. However, future high waves of SARS-CoV-2 and influenza may occur simultaneously, leading to multiple co-infections. Given that COVID-19 and influenza affect similar high-risk groups, co-infections of SARS-CoV-2 and influenza viruses may pose a serious threat, with implications for public health [9]. It was not until the later years of the pandemic that more aggregated data were published defining the role of such infections during the peaks and troughs in confirmed SARS-CoV-2 cases [10,11]. Globally, the incidence of such mixed infections between seasonal respiratory viruses and SARS-CoV-2 during the pandemic ranged from 3% to 8% [12,13,14,15]. However, data on the prevalence and role of coinfections between SARS-CoV-2 and other respiratory viruses are limited [16,17,18,19]. In our previous study, we elucidated the clinical role of mixed respiratory viral infections in SARS-CoV-2-positive patients [20]. Research on the genetic changes and mutations of respiratory viruses involved in such mixed infections is lacking, opening up unexplored opportunities for genetic variation. On the other hand, 6 million whole-genome sequences (WGS) of SARS-CoV-2 have been generated in EpiCoV by the Global Initiative for sharing of all influenza data (GISAID) [21]. This makes it possible to accurately calculate the evolution and transmission of SARS-CoV-2 [22,23]. During the approximately three-year-long COVID-19 pandemic, SARS-CoV-2 underwent various mutations and recombinations, leading to the occurrence of new variants with altered properties. In September 2020, a new variant of SARS-CoV-2, named Alpha (B.1.1.7), emerged in the UK [24]. The Delta variant (B.1.617.2) appeared in October 2020 in India and led to widespread distribution in 2021 [25]. Subsequently, a new variant, Omicron (B.1.1.529), emerged in the temperate latitudes of the Northern Hemisphere in December 2021 and caused the largest wave of COVID-19 in the winter and spring of 2022 [25]. The Omicron sub-variants BA-1, BA.2, BA.3, BA.4, BA.5, BQ1, and XBB successively emerged and dominated [26]. In 2023, due to the occurrence of multiple mutations in the genome of the XBB line of Omicron, several sublines developed, namely, XBB.1, XBB.1.5., and XBB.1.6.8. [27,28,29]. In September 2023, both XBB.1.5-like and Omicron line BA.2.86 circulated. These variants have amino acid substitutions that enable the virus to evade neutralizing antibodies [30]. As of the end of 2023, the JN.1 variant is the most prevalent across the world. Despite the extensive number of genetic studies conducted on SARS-CoV-2, studies aimed at elucidating the mutational activity of non-coronaviruses involved in mixed infections with SARS-CoV-2 are lacking or underrepresented [31]. In-depth genetic studies of co-infecting pathogens will contribute to a better understanding of their mutational activity. This will allow researchers to elucidate the mechanisms underlying the emergence of new variants and their possible recombinations. Our team set out to analyze the genetic characteristics of pathogens in cases of viral–viral mixed infections in patients with COVID-19 using a multiplex next-generation sequencing (NGS) approach. For more in-depth knowledge, we aimed to conduct an extensive genetic analysis to determine the genotypic identity of these co-pathogens and identify the amino acid substitutions in viral proteins that are associated with changes in the phenotypic properties of viruses.

## 2. Materials and Methods

### 2.1. Study Population and Specimen Sampling

Between October 2020 and the end of January 2024, nasopharyngeal samples were collected from 2277 SARS-CoV-2 positive patients in all 28 regions of Bulgaria, including outpatients and hospitalized persons. Patients of all ages were studied, ranging from 10 days to 98 years (mean age 53 ± 30.03 years; median age 67 years). Regarding the gender of the patients, the majority were female, 54%, while the proportion of male patients was 46%.

Swabs with media labeled by the manufacturer as suitable for viruses were used. Samples were transported to a national laboratory under refrigerated conditions at 4 °C. Before shipment, samples were frozen at 4 °C for up to 72 h before being shipped to the laboratory. Processing was performed on the day samples arrived at the laboratory, or samples were stored at −80 °C prior to testing.

### 2.2. Methodology for the Detection of Respiratory Co-Pathogens

Extraction

First, the samples were extracted using an automated extraction system with the ExiPrep Dx Viral DNA/RNA kit from Bioneer, Daejeon, Republic of Korea. The process took 1 h and 34 min, and the device allowed the selection of a program that could simultaneously isolate DNA and RNA in a single eluate. The final volume of the eluate was set to be 100 µL, considering the need to study a large number of pathogens.

Multiplex real-time polymerase chain reaction (PCR)

FluSC2 primers and probes donated by CDC (Atlanta, GA, USA) and a multiplex PCR system with Applied Biosystems™ TaqMan™ Multiplex Master Mix procured from Thermo Fisher Scientific (Waltham, MA, USA) were used for the simultaneous detection of SARS-CoV-2 and influenza A/B viruses.

All SARS-CoV-2-positive samples were then tested for other seasonal respiratory viruses, including respiratory syncytial virus (RSV), human metapneumovirus (HMPV), parainfluenza virus (PIV) types 1/2/3, rhinovirus (RV), adenovirus (AdV), and bocavirus (BoV). Screening for non-influenza respiratory viruses was performed using a multiplex PCR system with Applied Biosystems™ TaqMan™ Multiplex Master Mix (Thermo Fisher Scientific, Waltham, MA, USA). Different probes labeled with the following fluorescent dyes were used: FAM, HEX, Texas Red, and Cy5.

Three real-time PCR mixes were prepared to detect multiple virus combinations simultaneously:

Mix 1: AdV-Texas Red + RSV-HEX + PIV-FAM.

Mix 2: BoV-Cy5 + RV-HEX + PIV2-FAM.

Mix 3: HMPV-HEX + PIV3-FAM.

The following temperature conditions were used to detect SARS-CoV-2, influenza, and other respiratory viruses:

Reverse transcription: 25 °C for 2 min and then 50 °C for 15 min;

Initial denaturation: 95 °C for 2 min;

Amplification for 45 cycles: 95 °C for 15 sec and then 55 °C for 30 sec.

A QuantStudio™ 5 96-well real-time PCR system from Thermo Fisher Scientific (Waltham, MA, USA) was used to test for all respiratory viruses. Samples with a cycle threshold (Ct) value < 38 were considered positive. Samples that tested positive for SARS-CoV-2 and other respiratory viruses with a Ct value below 31 were selected for NGS. The samples were re-extracted following a previously described protocol, and the Ct value was re-measured using real-time PCR.

### 2.3. Sequencing Methodology and Data Analysis

Multiplex Next-Generation Sequencing

The method of targeted multiplex NGS using a respiratory virus panel kit was performed for the simultaneous isolation of the genome of viruses involved in mixed infections. Illumina RNA Prep with Enrichment (L) Tag (Illumina, San Diego, CA, USA) was used to characterize 40 common respiratory viruses. Metagenomic sequencing was performed using the Illumina MiSeq system with the reagent kit v3 for 150 cycles (Illumina, San Diego, CA, USA). DNA libraries were analyzed for fragment size distribution using QIAxcel Advanced capillary electrophoresis (Qiagen, Hilden, Germany). Normalization of the libraries was performed with a Qubit 4 Fluorometric and Invitrogen™ Quant-iT™ 1X High-Sensitivity (HS) Broad-Range (BR) dsDNA Assay Kit (Invitrogen, Thermo Fisher Scientific, Waltham, MA, USA).

Genome, phylogenetic, and amino acid analyses

We used Explify RPIP Data Analysis software (v2.0.0) available on the BaseSpace platform (Illumina, Cambridge, UK) for genome analysis. This software also analyzes the presence of resistance to influenza neuraminidase and endonuclease inhibitors. The genetic sequences of the detected respiratory viruses were deposited in the GenBank and GISAID sequence databases. The accession numbers of the deposited sequences are presented in Appendix A. Variant assignment analysis of SARS-CoV-2 was performed using the Pangolin COVID-19 Lineage Assigner v4.3 program (https://pangolin.cog-uk.io, 12 June 2024).

BLAST searches were performed in multiple databases, including GenBank, GISAID Epiflu, EpiCoV, and EpiRSV, by retrieving references and closely related sequences. Geneious Prime (GraphPad Software, LLC, Boston, MA, USA), software was used for alignment and phylogenetic tree construction, and the Interactive Tree Of Life (iTOL) software (https://itol.embl.de, 12 June 2024) was used for the overall design of the trees.

Amino acid analysis was performed with BoEdit (www.mbio.ncsu.edu/BioEdit/BioEdit.html, 12 June 2024) (RRID: SCR_007361) using the consensus sequences listed in Appendix A. The variable regions of the amino acid sequences were subjected to glycosylation site analysis. The NetNGlyc 1.0 and 4.03 servers seeded N-glycosylation sites and predicted GalNAc O-glycosylation sites. Sites with scores greater than the threshold of 0.5 were classified as being glycosylated.

### 2.4. Statistical Analysis

Categorical variables were analyzed using Chi-square or Fisher’s exact tests, and the results were presented as total counts and percentages. Continuous variables were compared using various statistical tests, including paired comparison plots, Fisher’s least significant difference (LSD) tests, *t*-tests, and Mann–Whitney U-tests. Any observed differences with *p*-values less than 0.05 were considered statistically significant. The analysis was conducted using Origin, a data analysis and graphing software package.

## 3. Results

### 3.1. Patient Characteristics

Between October 2020 and January 2024, a total of 2277 samples were collected from patients who tested positive for SARS-CoV-2. These samples were examined to determine the presence of coinfections with influenza A/B and eight other respiratory viruses. This study included patients of different age groups: 527 (22.3) were under 16 years of age, 473 (20.7%) were between 17 and 64 years of age, and 1158 (50.9%) were over 65 years of age. The age range of the patients was from 45 days to 98 years, with a mean age of 52 ± 30.6 years and a median age of 57 years. Mixed infections involving SARS-CoV-2 were detected in 164 (7.2%) patient samples.

### 3.2. Seasonal and Age Distribution of Cases with Proven Coinfections

An analysis of the seasonal distribution of the detected cases with mixed infections revealed the highest rate (6.9%, 62/888) of coinfections involving SARS-CoV-2 in 2021, followed by 2022 (6.8%, 35/517) and 2023 (5.8%, 43/743). Because of a relatively shorter winter period at the end of 2020 and at the beginning of 2024, we recorded a peak of coinfections involving SARS-CoV-2 of 7.8% (4/51) and 23% (19/73), respectively. From February to December of 2021, 2022, and 2023, a gradual rise in coinfections involving SARS-CoV-2 and other respiratory viruses was observed. In both March and June of 2021 and 2022, spikes in confirmed coinfections were registered. The peak of mixed infections in 2021 was caused by HMPV (18.2%), while in 2022, it was caused by RSV (50%). In 2023, the peak of these coinfections shifted to May and August (15.4% and 11.5%, respectively). In January 2024, the peak of coinfections involving SARS-CoV-2 (27.9%) was caused by an increased number of confirmed mixed infections with influenza A viruses (15.3%) (Figure 1).

The highest percentage (15%) of coinfections was found among children and adolescents <16 years of age, followed by adults aged 17–64 years (6%) and older adults aged 65+ years (4%) (*p* = 0.0001). The group of patients under 16 years old had a significantly higher rate of mixed RV and BoV infections than the other two age groups (*p* < 0.05). No coinfections between influenza and SARS-CoV-2 viruses were found in individuals aged 65 or older (Table 1).

### 3.3. Viral Load in Mixed Infections

We compared the viral load of the participants with mixed infections, as expressed indirectly by the cycle threshold (Ct). A comparison of the mean Ct value between SARS-CoV-2 and other non-influenza respiratory viruses showed that the viral load of SARS-CoV-2 (24.2 ± 5.3) was higher than that of the other co-pathogens (33.4 ± 4.9) (*p* < 0.001) (Figure 2A). The mean Ct value (26.2 ± 3.7) was lower in mixed infections of influenza viruses with SARS-CoV-2 compared to mixed infections of SARS-CoV-2 with other respiratory viruses (33.4 ± 4.9) (*p* < 0.001) (Figure 2B). When comparing the mean Ct values of SARS-CoV-2 and influenza viruses in mixed infections, both co-pathogens exhibited a high viral load (mean Ct = 26.2 ± 3.7 vs. 27.8 ± 3.9) (Figure 2C). We also found a significant difference in the mean Ct value of mixed infections of SARS-CoV-2 with influenza A/B viruses (24.2 ± 5.3) compared to that of SARS-CoV-2 and other non-influenza respiratory viruses (27.8 ± 3.9) (*p* < 0.01)) (Figure 2D).

### 3.4. Sequencing of Respiratory Viral Co-Pathogens

Thirty-six samples with detected co-pathogens with a Ct value < 28 were subjected to targeted NGS according to the instructions of a commercial kit. Among them, 21 (58.3%) of the sequenced samples were from male patients and 15 (41.7%) were from female patients (Table 2).

Twenty-three patient samples with mixed infections were successfully sequenced to determine SARS-CoV-2 variants. The distribution of respiratory viral co-pathogens relative to a given SARS-CoV-2 variant was analyzed (Table 3). The Alpha variant was found in 4 (17.4%) samples, the Delta variant in 8 samples (34.8%), and the Omicron variants in 11 samples (47.8%). In the remaining 13 samples, the sequences of one or both viral participants were not of sufficient quality to undertake further analyses.

### 3.5. Phylogenetic Analysis and Amino Acid Analyses of Influenza Viruses Involved in Mixed Infections with SARS-CoV-2

Complete influenza virus genomes were isolated simultaneously with that of SARS-CoV-2 in six patient samples—three with A(H1N1)pdm09, two with A(H3N2), and one of the B/Victoria lineage.

A phylogenetic analysis was performed to determine the affiliation of the Bulgarian sequences to the globally distributed genetic groups of influenza viruses. Two influenza A(H1N1)pdm09 viruses belonged to clade 6B.1A.5a.2a and one virus belonged to clade 6B.1A.5a.2a.1 (Figure 3a). The influenza A(H3N2) sequences fell into the genetic groups 2a.1b and 2b within the 3C.2a1b.2a.2 clade (Figure 3b). The influenza B/Victoria lineage belonged to the genetic clade V1A.3a.2 (Figure 3c).

The amino acid sequences of the influenza viruses detected in a coinfection with SARS-CoV-2 were compared with those of the World Health Organization (WHO) Northern Hemisphere vaccine strains to identify substitutions that might affect vaccine effectiveness.

The amino acid substitutions found in the Bulgarian sequences of A(H1N1)pdm09 viruses compared to the vaccine strain A/Victoria/2570/2019 are presented in Table 4. A total of 30 amino acid changes were identified in the HA protein, including R142K and S157T at the antigenic sites Sa and Ca2, as well as 13 substitutions in the NA protein. In addition, two R223Q mutations, which play a role in altering host specificity, were observed in all three sequences. The R223Q substitution is at a host cell receptor recognition site [32]. New D50N glycosylation sites were observed in all three NA protein sequences, which may also affect these strains’ antigenic and other properties.

We observed N-glycosylation sites at the HA1 positions 10, 23, 87, 162, 276, and 287 and the HA2 positions 154 and 213. All of these sites were present in the vaccine strain. All three Bulgarian co-pathogens contain eight N-glycosylation motifs in NA (positions 42, 50, 58, 63, 68, 88, 146, and 235).

The HA and NA amino acid sequences of two co-infecting influenza A(H3N2) viruses were compared with those of the A/Darwin/9/2021 vaccine virus using FluSurver (Table 5). The substitution N202D, located at the receptor-binding site of the HA protein, was found in all A(H2N3) sequences. The HA substitution G225D observed in both Bulgarian strains was responsible for the change in host specificity. Two HA1 substitutions I140K and S156H affect antigenic sites and are associated with antigenic drift/escape mutation. An analysis of the NA protein revealed the presence of D463N substitution, which leads to the loss of a potential N-glycosylation site. This mutation was found in 118 samples in three countries (which is 5.92% of all samples with the NA sequence [33]).

N-glycosylation motifs at the HA1 positions 8, 22, 38, 45, 63, 122, 126, 133, 165, 246, and 285 and the HA2 position 154 were detected. Nine potential N-glycosylation motifs were detected in NA (61, 70, 86, 146, 200, 234, 245, 367, and 463).

The HA and NA sequence identity of the B/Victoria-lineage virus detected in a coinfection with SARS-CoV-2 was 97% and 99.1%, respectively, compared to the vaccine strain B/Austria/1358417/2021. The HA protein harbored 23 substitutions compared to the vaccine strain, namely, G71H, P73S, K74G, T76S, G77del, K78del, I79L, S81A, V93I, G154D, F155del, F156del, A157S, T158S, M159Q, A160Q, A162L, V163G, ins164S, K165Q, D197E, A202del, and R203K. Four mutations, including I79L, V163G, ins164S, K165Q, and R203K, were located in antibody-binding regions. The substitution at position 163 affected the receptor-binding site, while those at positions 164 and 165 were associated with increased virulence. The R203K substitution was located at the host cell receptor-binding site and was involved in a T-cell epitope presented by the major histocompatibility complex (MHC) molecule. Three substitutions were found in the NA protein of the B/Victoria-lineage viral sequence compared to the vaccine strain, namely, G378E, Q453K, and I459V.

Eleven putative N-glycosylation motifs were identified at HA1 positions 25, 59, 145, 166, 233, 304, and 333 and at HA2 positions 145, 171, 184, and 216 (145, 166, and 197 fall into the following antigenic sites: 150-loop, 160-loop, and 190-helix). Four putative N-glycosylation motifs were identified in NA at positions 56, 64, 144, and 284.

### 3.6. Genetic Analysis of RSV-B Involved in SARS-CoV-2 Mixed Infections

The phylogenetic analysis revealed that all 10 RSV-B sequences detected in co-infection with SARS-CoV-2 belonged to the genetic group GB5.0.5a (Figure 4) joining algorithm in Geneious Tree Builder. The sequences of the reference strains representative of the known genotypes were obtained from GenBank with the corresponding ID numbers. The tree was rooted based on the sequence deposited by a research team in USA in 1991 (WV10010/USA/1991/M73541/GB1). Isolated RSV-B sequences from patients infected with SARS-CoV-2 are highlighted in yellow.

The G-protein sequences of RSV-B isolates were compared with that of the reference GB5.0.5a. strain hRSV/B/Australia/VIC-RCH056/2019. In total, 30 amino acid substitutions were identified in the G protein of the isolates, with an average number of 13.2 (Table 6).

The F protein of RSV-B was found to carry the amino acid substitutions S190N, N201S, S211N, and S389P when compared to the reference strain hRSV/B/Australia/VIC-RCH056/2019. The substitutions N201S and S211N were located in the immunodominant Ø epitope, which is present in the pre-fusion conformation of the F protein, and the substitution S389P was located in antigenic region I.

Five possible N-glycosylation sites were found in the G protein at positions 81, 86, 251, 256, and 294. Additionally, new glycosylation sites were found in two of the sequences at position 251 and in 10 sequences at position 256 when compared to the reference strain. In the F protein, possible N-glycosylation sites were found at five positions: 27, 70, 116, 126, and 500. No new N-glycosylation sites were identified when compared to the reference genome hRSV/B/Australia/VIC-RCH056/2019.

The N protein of all 10 RSV-B sequences carried the V97I substitution when compared to the reference strain N hRSV/B/Australia/VIC-RCH056/2019. This substitution is implicated in host cell-binding sites and viral oligomerization interfaces.

### 3.7. Genetic Analysis of AdV Involved in Mixed Infection with SARS-CoV-2

Based on the phylogenetic analysis, both AdV sequences involved in mixed infections with SARS-CoV-2 belonged to HAdV-C type 1 (Figure 5). The HAdV-C1 sequences were most closely related to an isolate from the 2009 USA collection (HAdV-C1/USA/3P2/2009-OQ518280.1).

An extensive amino acid analysis of the surface proteins—hexon, fiber, and penton (III)—was performed, and the results were compared to the reference strain AC_000017.1. No amino acid substitutions were detected in the hexon protein, and a single amino acid substitution E301D was identified in the penton protein near region 340 of the RGD loop (n = 1/574, 0.2%) (Figure 6). A higher number of amino acid substitutions was observed in the fiber protein compared to the other two regions, indicating greater variability (n = 15/582, 2.7%) (*p* = 0.0005 vs. penton protein). These amino acid substitutions included A71T, K74E, N199S, D246Y, R266L, R267V, L268V, H339N, H414Y, R442K, F461L, N462Y, E463K, S472G, and E527D.

N-glycosylation sites were detected in the hexon, fiber, and penton proteins. The highest percentage of glycosylated sites (2.2%) was found in the fiber protein, where 13 out of 582 sites were glycosylated at positions 78, 83, 97, 171, 186, 295, 349, 394, 470, 488, 538, 543, and 562 (*p* = 0.005 vs. hexon; *p* = 0.02 vs. penton). In the hexon protein, five glycosylation motifs were identified (0.5% out of 965) at positions 48, 353, 451, 838, and 963. In the penton protein, only three glycosylation sites (0.5% out of 574) were detected at positions 78, 175, and 312.

We conducted an analysis to identify the potential targets for O-glycosylation in the three abovementioned proteins. The highest percentage of serine and threonine configuration belonging to the CARBON groups was found in the hexon protein (n = 13/110, representing 11.8%). Potential sites for O-glycosylation with a score greater than 0.5 were located at positions 49, 53, 58, 112, 239, 240, 554, 611, 613, 636, 659, and 796.

When compared to the reference strain, we found a different O-glycosylation site at position 58 in the Bulgarian strains. Additionally, the reference strain lacked three additional O-glycosylation sites at positions 197, 211, and 507 that we detected in the Bulgarian strains.

In the fiber protein, 125 sites in the CARBON groups were identified in the Bulgarian strains. Potential O-glycosylation sites were found at positions 35, 84, 87, 101, 113, 116, 143, 161, 176, 177, 178, 182, and 404. Three new O-glycosylation sites at positions 87, 113, and 143 were detected, which were not present in the reference strain. On the other hand, two O-glycosylation sites at positions 7, 80, 168, and 183 were identified in the reference strain but were not present in the Bulgarian strains.

In the penton protein, we identified 84 sites in the CARBON groups that were possible sites for O-glycosylation, with a score above 0.5 for nine sites (10.7%) at positions 20, 49, 295, 313, 321, 343, 346, 456, and 520. One new potential O-glycosylation site at position 456 was identified, while the remaining positions matched those of the reference strain.

### 3.8. Genetic Analysis of HBoV Involved in Mixed Infection with SARS-CoV-2

The phylogenetic analysis showed that both HBoV sequences obtained from the respiratory samples of patients infected with SARS-CoV-2 were closely related to the sequence of the prototype HBoV strain St1 (Figure 7). HBoV-V1/Bulgaria/1305/2023 exhibited a nucleotide similarity of 99.7% with another Bulgarian sequence identified in 2017. In this study, the nucleotide similarity between HBoV-V1/Bulgaria/275/2021 and HBoV-V1/Bulgaria/1305/2023 was 97.6%.

An analysis of the amino acid composition of the VP1/VP2 protein in the two HBoV sequences we isolated, relative to the reference strain St1, revealed the presence of the A20T substitution. The isolate HBoV/Bulgaria/275/2021 had an additional T461S substitution when compared to the reference strain (Figure 8).

Seven putative N-glycosylation sites at positions 20, 82, 168, 211, 279, and 391 were identified in the VP1/VP2 protein, which accounted for 1.3% of the total sites. O-glycosylation in this protein was also analyzed, and 10 out of 81 CARBON elements with scores greater than 0.5 were found, indicating possible glycosylation sites. These sites were located at positions 2, 4, 13, 20, 21, 25, 27, 281, 292, and 362. In comparison to the prototype strain, two new O-glycosylation sites at positions 2 and 2 were detected in the Bulgarian sequences.

### 3.9. Genetic Analysis of PIV3 Involved in Mixed Infection with SARS-CoV-2

The phylogenetic analysis showed that the PIV3 strain detected in cases with mixed infection with SARS-CoV-2 belonged to cluster 1 subgroup “a” and genetic line 1a(i). This strain was highly genetically related to a 2017 Indian mock collection, with a nucleotide similarity of 99.3% (MH330335) (Figure 9).

An amino acid analysis of the fusion (F) protein and hemagglutinin–neuraminidase protein (HN) of the PIV3 sequence was performed. The F-protein analysis revealed the presence of multiple amino acid substitutions, namely, P2L, T3I, L7F, F17H, S95A, K108E, I165V, V367T, N401D, N441S, N486S, V497I, I499T, I501M, V509I, I513T, V516I, I521L, when compared to the reference strain S82195.1. The following mutations were found in the HN protein when compared to the reference strain JN089924.1: M21T, A22T, I28L, I53T, H62R, D68N, V69I, F73L, I87T, M118I, H295Y, and I391V. According to the analysis, the proportion of detected substitutions was found to be higher in the F protein (3.3% or 18/539) than in the HN protein (2.1% or 12/572) (Figure 10).

Four N-glycosylation sites at positions 239, 360, 447, and 509 were identified in the F protein and two were identified at positions 352 and 524 in the HN protein with a frequency of 0.7% and 0.3%, respectively. Overall, 2 of 84 (2.3%) and 5 of 103 CARBON groups (4.9%) were potentially O-glycosylated in the F protein and the HN protein, respectively. The F protein had O-glycosylation sites at positions 244 and 245, which corresponded to the same sites in the reference strain. Meanwhile, the HN protein had O-glycosylation sites at positions 143, 352, 353, and 359, which also corresponded to the same sites in the reference strain. Furthermore, we found a new O-glycosylation site at position 126 in the HN protein that was not present in the reference strain.

## 4. Discussion

Many studies conducted during the COVID-19 pandemic reported possible coinfections of SARS-CoV-2 with other respiratory pathogens, including bacteria, fungi, and viruses [12,34,35]. In our three-year follow-up study, a rate of 7.2% of coinfections was detected, which is consistent with other reports [4,36,37]. Seven types of seasonal respiratory viruses were identified in mixed infections with SARS-CoV-2, among which RSV was the most common co-pathogen. A study conducted in Italy between 2021 and 2022 also reported a high prevalence of RSV coinfection with SARS-CoV-2 in children [38]. Similarly, a 2020–2021 study in Ecuador found RSV to be the second most common co-pathogen after influenza A [35]. Much of the research conducted early in the pandemic reported the presence of coinfections [14,34,39]. Other studies examined the clinical burden of such mixed infections [20,37,40]. Studies examining the genetic characteristics of respiratory co-pathogens involved in mixed infections with SARS-CoV-2 are so far absent or are very limited in the literature. To address this research gap, our study tracked the incidence, seasonality, and age distribution of such mixed infections. It also tracked the genetic characteristics of each respiratory co-pathogen and the possibility of mutations in its viral genome. This made our study unique, and the results are important for elucidating the role of respiratory viral coinfections with SARS-CoV-2.

The COVID-19 pandemic affected the prevalence, seasonality, and incidence of seasonal respiratory viral infections, which also had an impact on the epidemiology of coinfections [35]. At the beginning of the pandemic, strict anti-epidemic measures were implemented that resulted in a low rate of reported coinfections, even in countries with a high prevalence of SARS-CoV-2 [41]. Our study found a 7.8% co-infection rate in the winter quarter of 2020, which was higher than the value (6.9%) in 2021. This higher rate may be explained by the fact that surveillance occurred only in the autumn–winter months when respiratory infections were at their peak. Several studies indicated that non-pharmaceutical measures taken to reduce the spread of SARS-CoV-2 up until 2022 have also affected the prevalence of influenza viruses and RSV in terms of their seasonality and geographical distribution [42,43]. Consistent with a previous study that reported an absence or a low prevalence of influenza viruses in the 2020–2021 and 2021–2022 seasons [44], our study found low or no rates of coinfections with influenza viruses in these seasons. However, during the winter months of 2023–2024, we observed an increase in mixed infections, mainly due to mixed infections of SARS-CoV-2 with influenza A viruses (15%). This increase in coinfections compared to the previous two flu seasons was because, in the winter season of 2023–2024, the flu wave reached the same level as before the COVID-19 pandemic. Additionally, during the winter months, the number of patients infected with SARS-CoV-2 increased [45,46]. In our previous study, we reported an unusual shift in the seasonal peak of RSV during the summer months of 2022 [47]. We hypothesized that the higher percentage of individuals infected with RSV accounted for the peak of co-infections with both SARS-CoV-2 and RSV.

Based on the above findings, we established a seasonal regularity in the increase in mono-infections, which gives an impetus to the increase in cases of coinfections with SARS-CoV-2. We also established a relationship between the age of patients and the proportion of confirmed coinfections. According to our research, children and adolescents under 16 years of age were more prone to mixed infections with SARS-CoV-2. Previous studies have shown that this phenomenon is also common in children who tested negative for SARS-CoV-2 [48,49]. The prevalence rate of children with COVID-19 is also known to be lower than that of older patients [50,51,52,53]. This is explained by the fact that children are usually asymptomatic when infected with SARS-CoV-2 or have mild symptoms [54]. Children co-infected with SARS-CoV-2 exhibit more frequent clinical manifestations and symptoms than children with mono-infections, giving reasons for more frequent testing and detection of such mixed infections [4]. RV and BoV infections are more common in children and adolescents [55,56], which also explains the higher rate of coinfections found in SARS-CoV-2-positive children compared to the older age groups of 17–64 and over 65 years (*p* < 0.05). We did not detect mixed infections of influenza and SARS-CoV-2 in patients older than 65 years. Another study reported that in these patients, coinfections are very likely to result in the need for intensive care [57]. We observed similar average viral loads of SARS-CoV-2 and influenza viruses, allowing us to conclude that infection with these two viruses occurred in a close period. This is another reason for concern when such a coinfection occurs, which can complicate the disease and lead to death, especially among patients over 65 years of age with accompanying diseases [58]. In the other cases with coinfections, the viral loads of non-influenza respiratory co-pathogens were found to be significantly lower than that of SARS-CoV-2 (*p* < 0.001). An infection with a non-influenza respiratory pathogen was speculated to have developed earlier, followed by an infection with SARS-CoV-2, which suppressed the replication of the co-infecting virus.

While some authors reported cases of asymptomatic co-infected patients, other studies showed a more severe clinical picture in the presence of mixed infections in patients with COVID-19 [17,59]. This highlights the need to detect multiple respiratory pathogens simultaneously and determine the genetic characteristics of each co-pathogen. In this study, simultaneous detection of multiple respiratory co-pathogens was performed using both multiplex real-time PCR and multiplex NGS analysis. Several studies have also used multiplex sequencing analysis to detect the presence of co-pathogens [13,31,60,61,62]. By applying target (t) NGS analysis, we identified SARS-CoV-2 variants found in mixed infections similar to those reported in other studies using this approach [63,64,65]. In our study, a highest percentage of mixed infections was found with the Omicron variant SARS-CoV-2. This could be attributed to the gradual easing of safety measures after the emergence of this variant and its faster spread compared to previous variants [66]. In Bulgaria, the higher percentage of coinfections with the Delta variant compared to the Alpha variant was due to the larger wave of Delta infections [67].

Next-generation metagenomic sequencing (mNGS) has emerged as a efficient and accurate diagnostic strategy for detecting lower respiratory tract infections in recent years [68]. However, some authors have reported the need to confirm coinfections through PCR for the results to have a diagnostic value [69]. In this study, we implemented an alternative tNGS analysis, which was not used for diagnostic purposes. Instead, we used this method to track the genetic changes of some sequences isolated from respiratory co-pathogens found in cases with coinfections with SARS-CoV-2. It is crucial to analyze the evolutionary relationship of influenza viruses in different parts of the world to determine the effectiveness of vaccines [70]. The global circulation of the established clades of A(H1N1)pdm09 6B.1A.5a.2a and 6B.1A.5a.2a.1 is ongoing, but their relative proportions vary by region. In Europe, 5a.2a and 5a.2a.1 viruses are widespread, with 5a.2a viruses accounting for a larger proportion [71]. Five antigenic sites have been identified in the HA1 subunit of hemagglutinin: Sa, Sb, Ca1, Ca2, and Cb; amino acid substitutions in these regions can lead to a change in the antigenic properties of viruses [72]. In our study, two amino acid substitutions were identified at the antigenic sites Sa and Ca2. The N-linked glycan chains attached to the HA globular head restrict antibody access to these antigenic sites and enable viruses to escape humoral immunity [73]. Eight N-linked glycosylation sites were found in the HA protein of A(H1N1)pdm09 viruses. The R223Q mutation we identified in HA1 of HA also plays a role in restoring α-2,6 binding ability, as well as modestly reducing α-2,3 binding specificity [74]. The HA2 position 112, located in the long helix near the fusion peptide, is a mutation we found at position D112G/D439, which consistently leads to increased fusion pH. The HA2 position 112 contains several group-specific ionizable residues that have the potential to increase HA resistance [75]. The 2022–2023 winter influenza vaccine in Europe included the 3C.2a1b.2a.2a clade A (H3N2), which was also widespread in Europe during that season [76]. Several genetic clades of influenza A(H3N2) viruses circulated in Europe during the 2022–2023 season—2a.1, 2a.1b, 2b, and 2a.3a.1 [77]—and the Bulgarian strains belonged to the clades 2a.1b and 2b. Many amino acid substitutions were detected in the HA and NA glycoproteins of influenza A(H3N2) viruses, including C, B, and D at the antigenic sites. The substitution S156H affects the B-flanking RBS region. A previous study showed that the addition of the 158–160 glycan resulted in a significant antigenic shift during the winter season of 2014–2015 [78]. Antigenic drift/escape mutation was observed due to a substitution at the antigen recognition site of the NA segment of the Bulgarian strain N221G (Gulati U, 2002) [79].

The influenza B/Victoria virus identified in a co-infection with SARS-CoV-2 belonged to the genetic clade V1A.3a.2, which was most prevalent in Europe [76]. Since March 2020, no B/Yamagata-lineage viruses have been detected around the world [80,81]. In type B viruses, the major antigenic sites are located in the 120-loop, 150-loop, 160-loop, and 190-helix regions. The Bulgarian influenza B/Victoria sequence has deletions and substitutions that occur at the antibody-binding sites of HA, including G77del, K78del, and S81A, as well as V93I and A157S. The receptor-binding sites are located in the 140-loop, 190-helix, and 240-loop regions. In the B/Victoria sequence, substitutions have been found in the 160-loop and 190-helix regions [82]. According to literature data, the 342 mutation we found in the NA protein of influenza B viruses is located at an antigen recognition site, and the 453 mutation is at a site involved in host protein binding [83,84].

Both surface proteins of influenza A/B viruses (HA and NA) and those of RSV (G and F) have undergone evolutionary changes over time to evade host immune defenses [85,86]. This leads to the emergence of new genetic variants over time. According to the GISAID database, during the 2022–2023 season, the predominant RSV genotypes were GA2.3.5 for RSV-A and GB5.0.5a for RSV-B. The 10 RSV sequences isolated in this study in cases with coinfections with SARS-CoV-2 were type B and belonged to the GB5.0.5a genotype [87]. In previous research, six major antigenic sites have been identified in the RSV F protein: Ø, I, II, III, IV, and V [83]. In our study, two substitutions were found in the F protein of RSV-B: N201S in Ø and S389P in I. The confirmation of Ø before fusion and the preferential presence of I after fusion elicit specific antibodies. The Ø site has been determined to be the most variable, suggesting that antibodies binding to this site may be related to subgroup specificity [88]. This site is a targeting region for prophylactic antiviral therapy or a suitable site for vaccine development [89].

In AdVs, the most variable proteins are hexon, penton, and fiber [90]. We analyzed these proteins in the Bulgarian AdV strain isolated from cases with mixed infections with SARS-CoV-2. A high number of mutations was found in the fiber protein. In the penton monomer, the insertion domain contains an RGD chain, which is highly mobile and variable. The integrin-binding RGD motif facilitates binding between the integrin and the penton base [91]. It is important to know that recombinant events estimated based on the three main capsid genes—penton base, hexon, and fiber—are important for elucidating the genesis of new and emerging pathogenic AdVs [92]. The observation of emerging mutations in these target gene regions in AdVs has intrinsic significance given the emergence of AdV type 7 isolated from the lungs or respiratory system of deceased patients with severe pneumonia [93,94,95].

In the context of constantly occurring mutations leading to the appearance of new variants of known viruses, the first case of HBoV was recorded in 2005 in a respiratory sample obtained from a child [96]. HBoV-1 was reported to be detected primarily as a respiratory pathogen, while HBoV-2, HBoV-3, and HBoV-4 were mainly detected in fecal samples as potential gastrointestinal pathogens [97]. In our study, both genetic sequences of HBoV detected in cases with coinfections with SARS-CoV-2 were of the HBoV-1 type, which corresponded to the respiratory symptoms seen in these patients. Our previous three-year study targeting the VP1/VP2 protein of bocaviruses showed minimal sequence variation from the reference St1 strain [98]. In this study, in comparison with the same reference strain St1, we found the presence of only two substitutions and a low level of glycosylation of the capsid protein. Other studies have reported similar results regarding the VP1/VP2 protein [99,100,101] and amino acid substitutions in the order of 1 to 8 [102,103].

Unlike bocaviruses, parainfluenza viruses have been reported to have a high substitution rate for type 3 (4.2 × 10^−4^ subs/site/year). It is estimated that a new subgroup develops approximately every 50 years and a new class every 107 years [104,105]. In the literature, PIV-3 from cluster 1 is reported to predominate, and the PIV-3 co-pathogen isolated from cases with mixed infections with SARS-CoV-2 also belongs to this cluster. A newly emerging strain is 1b(ii), which was identified via genome-wide analysis [106]. The variant we identified belongs to the other subgroup 1a, which emerged earlier than 1b. A study conducted in China reported a higher number of N-glycosylated motifs in the F protein compared to the HN protein, whereas those predicted as possible O-glycosylation sites were reported to occur in a higher percentage in the HN protein compared to the F protein [107]. These findings are completely consistent with those found in this study. Amino acid substitutions in the F protein meta-antigens A, AB, B, and C have been found to neutralize resistance to MAbs [108]. In this study, despite the multiple substitutions detected in the F protein, no such substitutions were detected in the PIV-3 strain.

Respiratory viruses are easily transmitted and can spread quickly, causing local outbreaks, epidemics, and even pandemics [109]. Due to their small genome and rapid mutation rates, they are capable of causing annual epidemics. In the past, influenza viruses had caused some of the largest pandemics, including the Spanish flu of 1918, which resulted in millions of victims. The emergence of the “new” coronavirus in 2019 led to the unprecedented COVID-19 pandemic, affecting the entire world and causing millions of deaths [110]. There have been alarming reports of deaths and severe cases caused by AdV type 7, particularly in children and military personnel. These reports highlight the need to expand the scope of monitoring the genetic variability of viruses responsible for respiratory infections [111,112]. Our study can serve to advance the concept of simultaneous monitoring of genetic changes in several respiratory pathogens, especially in cases of mixed infections. Renewing and expanding the genetic pool of the existing databases through studies investigating the genetic characteristics of respiratory viruses will stimulate future research.

We should mention several limitations in this study. First, only a small proportion of confirmed coinfections was sequenced due to the low viral load in most cases with non-coronavirus respiratory viruses. Second, not all non-coronavirus respiratory co-pathogenic sequences were of sufficient quality for further analysis. Moreover, the number of co-pathogens included in the phylogenetic analysis was limited, which means we could not make any definite conclusions about the evolutionary affiliation of the strains distributed in Bulgaria during the study period. It is important to mention that the Ct value we used is an indirect method of viral load determination. A major drawback of this method is that when comparing viral loads, we could not account for identical amplification efficiency of primers.

## 5. Conclusions

In this three-year study, a relatively low proportion of respiratory viral coinfections was found in SARS-CoV-2-positive patients. We observed diversity in the etiology of such viral–viral mixed infections. Most of these coinfections were observed in children and adolescents under 16 years of age. We established the genotypic affiliation of seven types of respiratory co-pathogens that were isolated simultaneously with the genetic sequences of the Alpha, Delta, or Omicron variants of SARS-CoV-2. We observed multiple substitutions, with some occurring at antibody-binding sites, and identified multiple glycosylation sites. This study, which aimed to conduct simultaneous characterization of genetic changes in co-infecting viruses, will help to expand the scope of tracking the mutational ability of respiratory pathogens. It will also help clarify the mechanism underlying mutations through which such pathogens skillfully bypass the already-acquired immunity. The advance in genetic knowledge of co-pathogens will contribute to the improvement in vaccine effectiveness and the development of combination vaccines.

## Figures and Tables

**Figure 1 viruses-16-00958-f001:**
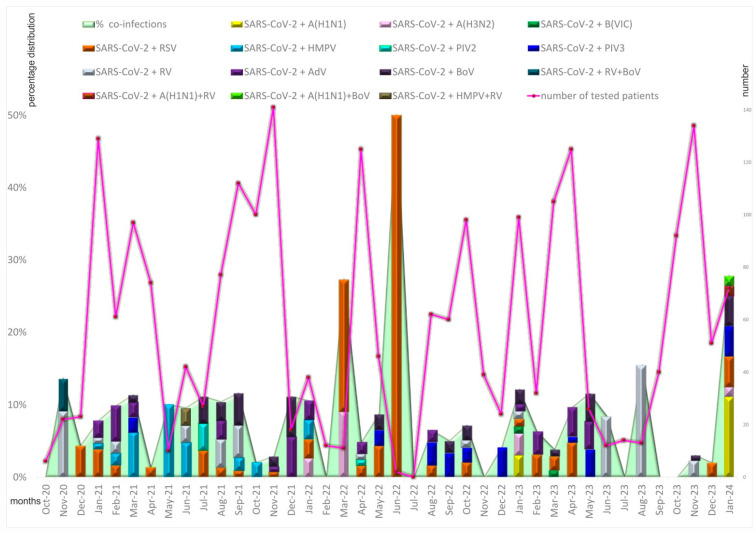
Distribution of proven coinfections by month from October 2020 to January 2024 in Bulgaria.

**Figure 2 viruses-16-00958-f002:**
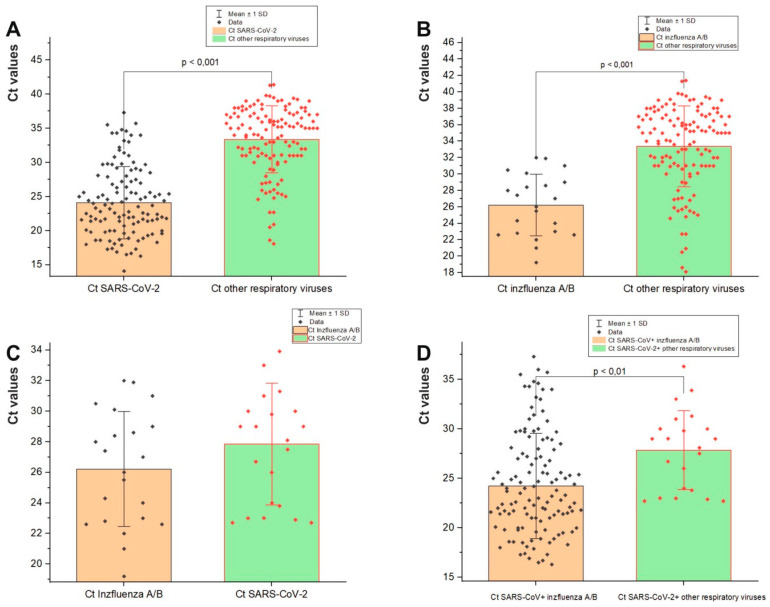
Comparison of Ct values of different respiratory viruses involved in mixed infections: (**A**) Ct of SARS-CoV-2 versus other respiratory viruses in a mixed infection (RSV, RV, BoV, HMPV, PIV2,3, and AdV); (**B**) Ct of influenza A/B viruses compared to Ct of other respiratory viruses involved in co-infection with SARS-CoV-2; (**C**) Ct of SARS-CoV-2 versus that of influenza A/B viruses involved in a co-infection; (**D**) Ct of SARS-CoV-2 co-infected with influenza A/B viruses versus Ct of SARS-CoV-2 in a mixed infection with RSV, RV, BoV, HMPV, PIV2/3, and AdV. Mean and median Ct values, SD, and *p* values (Fisher’s test) are shown. Values were calculated using the Mann–Whitney U-test.

**Figure 3 viruses-16-00958-f003:**
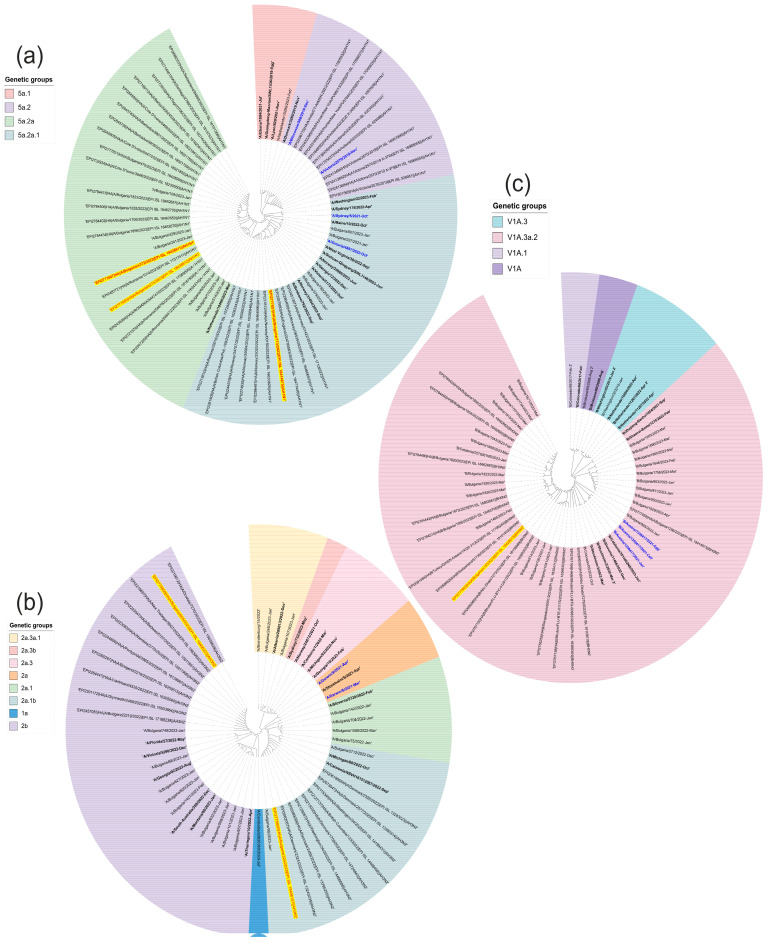
Phylogenetic analysis was performed to construct evolutionary trees based on a fragment of the HA gene of the influenza A(H1N1) virus (**a**). The Bulgarian sequences isolated in 2023 that belong to the influenza virus subtype A(H1N1) and that were detected in a co-infection with SARS-CoV-2 are marked in yellow. The tree was rooted based on the sequence deposited by a research team in Ghana in 2021 (A/Ghana/1894/2021-July). (**b**) Phylogenetic analysis of influenza virus A(H3N2). The Bulgarian sequences isolated in 2023 that belong to the influenza virus subtype A(H3N2) and that were detected in a mixed infection with SARS-CoV-2 are highlighted in yellow. The tree was rooted based on the sequence deposited by a research team in Brandenburg, Germany in 2022 (A/Brandenburg/15/2022). (**c**) Phylogenetic analysis of influenza virus B-Victoria. The sequences isolated in 2023 that belong to the B-Victoria influenza virus lineage and that were found in a mixed infection with SARS-CoV-2 are highlighted in yellow. The tree was rooted based on the sequence deposited by a research team in Colorado, USA in 2017 (B/Colorado/06/2 February 2017). Genetic distances were measured according to the Jukes–Cantor model. A phylogenetic tree was constructed using a neighbor-joining algorithm in Geneious Tree Builder. The sequences of the reference strains representative of the known genotypes were obtained from GenBank with the corresponding identification numbers.

**Figure 4 viruses-16-00958-f004:**
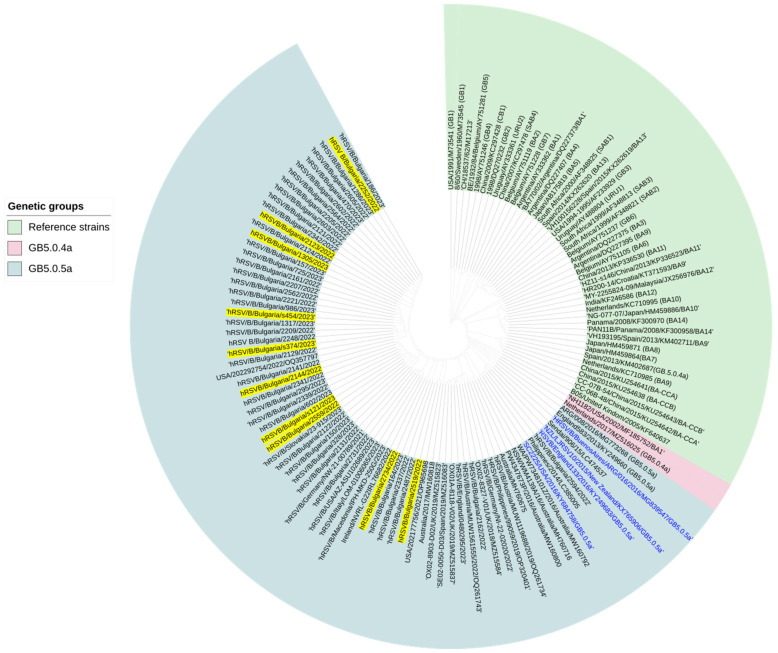
Phylogenetic analysis of RSV-B in which a tree was constructed relative to the G protein and the entire genome of the virus. Genetic distances were measured according to the Jukes–Cantor model. The phylogenetic tree was constructed using a neighbor.

**Figure 5 viruses-16-00958-f005:**
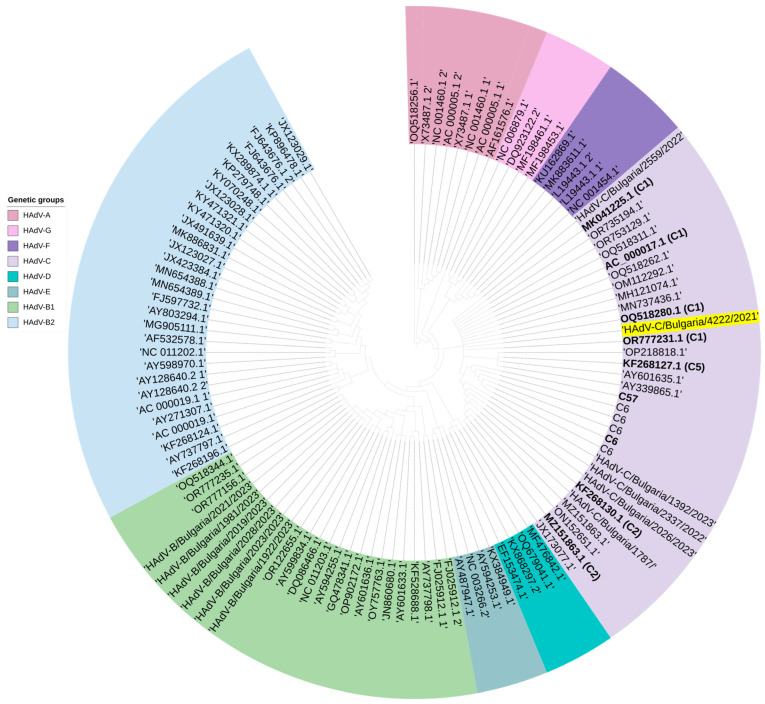
Phylogenetic analysis based on the hexon gene and the whole genome of HAdV. Genetic distances were measured according to the Jukes–Cantor model. The phylogenetic tree was constructed using a neighbor-joining algorithm in Geneious Tree Builder. The sequences of the reference strains representative of the known genotypes were obtained from GenBank with the corresponding identification numbers. The tree was rooted based on a 2010 strain (HAdV-A12 OQ518256.1). The sequence of HAdV-C1 isolated in mixed infections with SARS-CoV-2 is highlighted in yellow.

**Figure 6 viruses-16-00958-f006:**
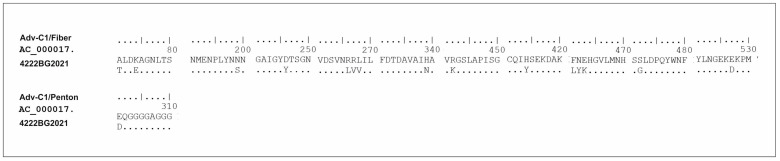
Amino acid analysis showing the sites of amino acid substitutions in the genetic sequence encoding the fiber and penton proteins of the Bulgarian strain of HAdV. Alignment was performed against the reference strain AC_000017.1.

**Figure 7 viruses-16-00958-f007:**
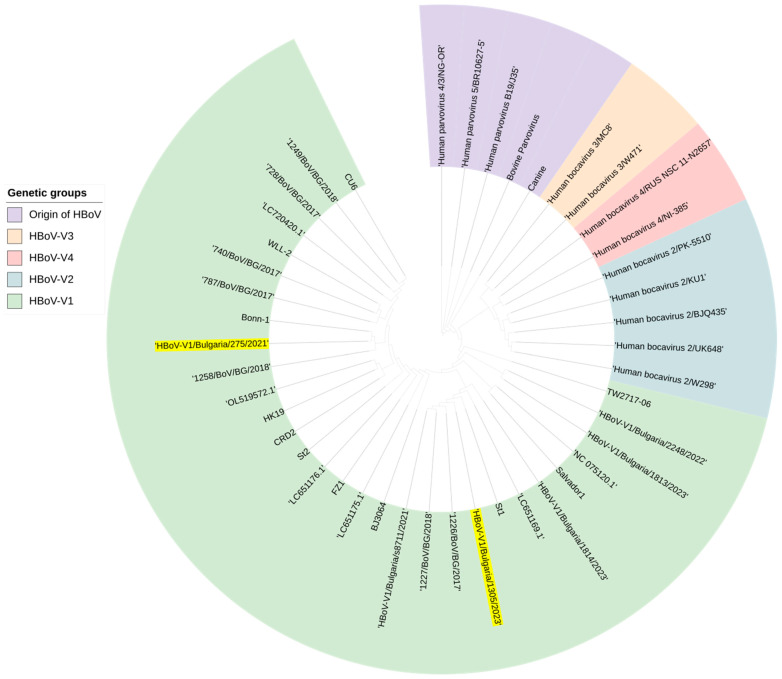
Phylogenetic tree constructed based on the VP1/VP2 gene and the whole genome of hBov and other parvoviruses. Genetic distances were measured according to the Jukes–Cantor model. The phylogenetic tree was constructed using a neighbor-joining algorithm in Geneious Tree Builder. The sequences of the reference strains representative of the known genotypes were obtained from GenBank with the corresponding identification numbers. The tree was rooted based on the strain Human_parvovirus_4/3/NG-OR. Isolated sequences from confirmed co-infected patients with HBoV-V1 and SARS-CoV-2 are highlighted in yellow.

**Figure 8 viruses-16-00958-f008:**
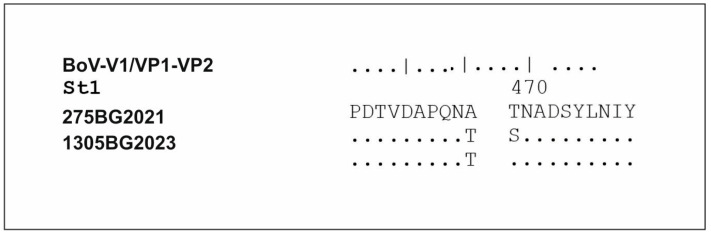
Amino acid analysis of Bulgarian strains isolated from cases with co-infection with SARS-CoV-2. Substitution sites in the genome of the Bulgarian HBoV strains encoding the VP1/VP2 protein are depicted. Alignment was performed against the prototype St1 strain.

**Figure 9 viruses-16-00958-f009:**
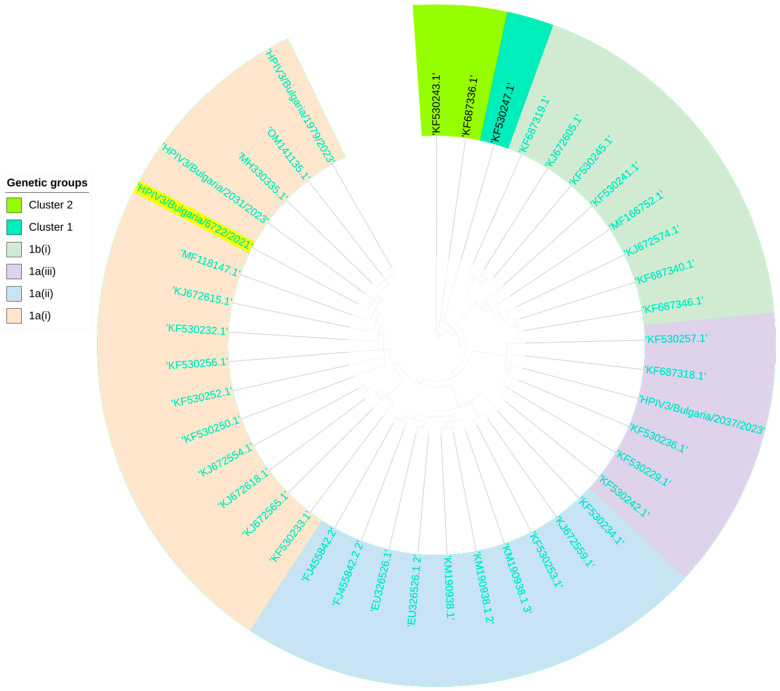
Phylogenetic analysis based on the HN-coding region and the whole genome of human parainfluenza virus type 3 (HPIV3). Genetic distances were measured according to the Jukes–Cantor model. The phylogenetic tree was constructed using a neighbor-joining algorithm in Geneious Tree Builder. The sequences of the reference strains representative of the known genotypes were obtained from GenBank with the corresponding identification numbers. The tree was rooted based on the clone 2 strain HPIV3/AUS/3/2007 (KF530243.1). PIV3 sequences isolated from SARS-CoV-2 co-infected patients and belonging to subclade 1a(i) are shown in yellow.

**Figure 10 viruses-16-00958-f010:**
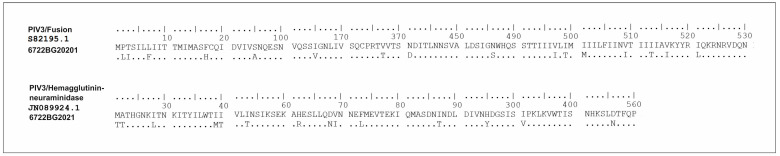
Amino acid analysis showing the sites of substitutions in the genetic sequences encoding the fusion (F) protein and hemagglutinin–neuraminidase protein (HN) of the Bulgarian strain of PIV3. Alignment was performed against the reference strains and S82195.1 and JN089924.1.

**Table 1 viruses-16-00958-t001:** Distribution of coinfections between SARS-CoV-2 and other respiratory viruses in different age groups (0–16, 17–64, and 65+) from October 2020 to January 2024.

Coinfections with SARS-CoV-2 and Other Respiratory Viruses											
AGE GROUPS	A(H1N1)	A(H3N2)	B-Victoria	RSV	RV	HMPV	PIV2	PIV3	BoV	AdV	Total
**0–16 years of age, ** **n = 527, n (%)**	7 (1.3)	6 (1.1)	0 (0)	16 (3)	16 (3)	5 (0.9)	1 (0.2)	3 (0.5)	20 (3.8)	7 (1.3.)	81 (15.4)
**17–64 years of age,** **n = 473, n (%)**	6 (1.3)	1 (0.5)	2 (0.4)	3 (0.6)	5 (1)	2 (0.4)	0 (0)	6 (1.3)	3 (0.6)	5 (1)	33 (6.9)
**>65 years of age,** **n = 1158, n (%)**	0 (0)	0 (0)	0 (0)	19 (1.6)	4 (0.3)	10 (0.9)	1 (0.1)	4 (0.3)	4 (0.3)	12 (1)	54 (4.6)
**Total coinfection cases,** **n = 2277, n (%)**	13 (0.6)	7 (0.3)	2 (0.08)	38 (1.7)	25 (1.1)	17 (0.7)	2 (0.08)	13 (0.6)	27 (1.2)	24 (1)	168 (7.4)

**Table 2 viruses-16-00958-t002:** Distribution of confirmed coinfections according to the number of viruses involved in the mixed infection.

	Two Co-Pathogens	Number of Coinfections	Three Co-Pathogens	Number of Coinfections	More Than Three Co-Pathogens	Number of Coinfections
Coinfections with the participation of SARS-CoV-2	A(H1N1)	3	A(H1N1) + RSV-B	3	RSV-B + AdV-C2 + BoV-V1	1
	A(H3N2)	2	A(H1N1) + HKU-1	1	RSV-B + HMPV + BoV-V1	1
	B-Victoria	2	A(H3N2) + RSV-B	2	A(H3N2) + RSV-B + PIV3 + AdV-C2 + BoV-V1	1
	RSV-A	1	RSV-B + AdV-E4	1		
	RSV-B	6	RSV-B + PIV3	1		
	PIV3-2	2	RSV-B + BoV-V1	1		
	AdV-C2	3	HMPV + AdV-C2	1		
	HPeV	1	HMPV + BoV-V1	1		
	BoV-V1	2				
Total number of coinfections		22		11		3

**Table 3 viruses-16-00958-t003:** Distribution of detected variants of SARS-Cov-2 and other respiratory viruses involved in mixed infections by date of sample collection.

Sampling Date	Variant	Pango Lineage	Co-Pathogens
2021-03-01	Alpha (B.1.1.7-like)	B.1.1.7	PIV3
2021-03-02	Alpha (B.1.1.7-like)	B.1.1.7	AdV-C2
2021-03-20	Alpha (B.1.1.7-like)	B.1.1.7	PIV3
2021-04-01	Alpha (B.1.1.7-like)	B.1.1.7	RSV-A
2021-07-12	Delta (B.1.617.2-like)	B.1.617.2	BoV-V1
2021-09-23	Delta (B.1.617.2-like)	AY.9.2	BoV-V1
2022-04-26	Omicron (BA.2-like)	A.2	AdV-C2
2022-11-10	Delta (B.1.617.2-like)	B.1.617.2	RSV-B
2022-11-10	Delta (B.1.617.2-like)	AY.92	RSV-B
2022-12-05	Delta (B.1.617.2-like)	B.1.617.2	A(H3N2) + RSV-B
2022-12-07	Delta (B.1.617.2-like)	AY.92	RSV-B
2022-12-15	Delta (B.1.617.2-like)	B.1.617.2	A(H3N2) + RSV-B + AdV-C2 + BoV-V2
2022-12-19	Delta (B.1.617.2-like)	B.1.617.2	RSV-B
2023-01-01	Omicron (BA.2-like)	CH.1.1	RSV-B
2023-01-06	Omicron (BA.5-like)	CK.1	B-Victoria
2023-01-11	Omicron (BA.5-like)	BA.5.1	A(H3N2)
2023-01-16	Omicron (BA.5-like)	EH.1	A(H1N1)
2023-01-16	Omicron (BA.5-like)	EH.1	A(H1N1)
2023-01-16	Omicron (Unassigned)	BQ.1.1.23	A(H3N2)
2023-02-22	Omicron (XBB.1.5-like)	XBB.1.5.20	A(H1N1)
2023-04-13	Omicron (XBB.1-like)	EG.1	A(H1N1)
2023-04-18	Omicron (XBB.1.5-like)	XBB.1.5	RSV-B
2023-05-02	Omicron (XBB.1.5-like)	EU.1.1	RSV-B

**Table 4 viruses-16-00958-t004:** Amino acid substitutions identified in the surface glycoproteins—hemagglutinin (HA) and neuraminidase (NA)—of three influenza A(H1N1)pdm09 viruses detected in a co-infection with SARS-CoV-2 compared to the vaccine virus A/Victoria/2570/2019 (without a signal peptide).

Alignment Against a Reference Genome in HA A/Victoria/4897/2022(H1N1)	Bulgarian A(H1N1) Sequences		
	HA_A/Bulgaria/s215/2023_18428912	HA_A/Bulgaria/s372/2023_1842891	HA_A/Bulgaria/773/2023_18414675
Clade	6B.1A.5a.2a (5a.2a)	6B.1A.5a.2a (5a.2a)	6B.1A.5a.2a.1 (5a.2a.1)
List of mutations/antigenic sites	S137P, R142K/Sa, A216T, R240Q, E260D, V272I, A277T, N356Y, L407Y, N408Y, H451N	S137P, R142K/Sa, A216T, R240Q, E260D, D269N V272I, A277T, N356Y, N431I, H451N	S157T/Ca2, A216T, R223Q, H438Q, D439G, S440I, N441stop, V442C, K443E, N444E, Y446V, E447stop, V449S, R450K, H451T, Q452S, L453V
% AA identity	98.1	98.2	97.3
**Alignment against a reference genome in NA A/Victoria/4897/2022 (H1N1)**	**NA_A/Bulgaria/s215/2023_18428912**	**NA_A/Bulgaria/s372/2023_18428911**	**NA_A/Bulgaria/773/2023_18414675**
List of mutations	I7K, I8T, N21S, D50N, S200N, E382G	D50N, S200N, E382G	I13V, D50N, S339L, E382G
% AA identity	99.1	99.3	99.1

**Table 5 viruses-16-00958-t005:** Amino acid substitutions identified in the surface glycoproteins—hemagglutinin (HA) and neuraminidase (NA)—of two influenza A(H3N2) viruses detected in a co-infection with SARS-CoV-2 compared to the vaccine strain HA A/Darwin/9/2021(H3N2) (without a signal peptide).

Alignment Against the HA of Vaccine virus A/Darwin/9/2021	Bulgarian A(H3N2) Sequences		
	HA_A/Bulgaria/325/2023_18406107	HA_A/Bulgaria/605/2023_18406223	
Clade	2a.1b	2b	
Amino acid substitutions/antigenic sites	D53G/C β, D104G/α, N186D/D G225D/*220loop*, K276R	E50I/C β F79V/α I140K/β S156H/B	N186D D G225D/*220loop* I242M L202V
% AA identity	99.1	98.6	
**Alignment against NA of vaccine virus A/Darwin/9/2021**	**NA_A/Bulgaria/325/2023_18406107**	**NA_A/Bulgaria/605/2023_18406223**	
Amino acid substitutions	T238A, D463N, N465S	V215I, D221N	G346S, S370T
		V263I	P386H
		R315S	E435K
% AA identity	99.4	99.3	

**Table 6 viruses-16-00958-t006:** Amino acid substitutions identified in the G protein of RSV-B involved in co-infection with SARS-CoV-2. The frequency of occurrence of the different isolates worldwide is shown.

Amino Acid Changes in the Bulgarian Strain	Frequency in the Bulgarian Strain, n (%)	Countries Where the Mutation Occurred (n)	Frequency of All Samples from Other Countries (%)
A74V	10 (100)	13	8.51
T131A	10 (100)	11	4.55
P214S	10 (100)	15	5.68
P221L	10 (100)	16	5.94
I252T	10 (100)	18	10.92
K256N	10 (100)	11	4.27
I268T	10 (100)	18	7.48
S275P	10 (100)	14	6.27
Y285H	10 (100)	20	8.68
I137T	9 (90)	11	4.25
S100G	8 (80)	9	2.87
P120L	2 (20)	NF *	NF
D251N	2 (20)	8	0.59
H277R	2 (20)	2	0.09
L71P	2 (20)	3	0.10
T225N	1 (10)	5	0.10
S309Y	1 (10)	NF	NF
T117A	1 (10)	1	0.02
S295T	1 (10)	1	0.02
T296H	1 (10)	1	0.02
Q297P	1 (10)	1	0.02
P299del	1 (10)	NF	NF
T292I	1 (10)	2	0.03
I59T	1 (10)	1	0.02
T139A	1 (10)	NF	NF
N143S	1 (10)	NF	NF
K145R	1 (10)	NF	NF
S147G	1 (10)	NF	NF
K153R	1 (10)	NF	NF
T209K	1 (10)	1	0.02

* NF = not found in sequences.

## Data Availability

This manuscript utilizes a database on the distribution of respiratory viruses in Bulgaria, which is accessible at https://grippe.gateway.bg/index.php, 12 June 2024.

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
