# Peer review of "Epidemiological and Genetic Characteristics of Respiratory Viral Coinfections with Different Variants of Severe Acute Respiratory Syndrome Coronavirus 2 (SARS-CoV-2)"

_viruses, 2024, doi:10.3390/v16060958_

Round 1

Reviewer 1 Report

Comments and Suggestions for Authors

The authors provide very good new epidemiological and genetic data on co-infections between known respiratory virus infections and Sars-Cov-2 with its different variants. It's an interesting article that provides a good basis for future studies.

I found very little to criticise overall. My only comment would be in the "Materials and methods" section, to create another small section and give more information about where the study was conducted, as well as the demographic characteristics of the participants such as sex, age, etc., for a better epidemiological understanding.

- In Line 84 : Give the informations concerning the medium used (company, country, year if possible)

- There seems to be a confusion in the title of Tabl S1 which is in the supplementary files (You have put Tabl S2).

- In Fig 4 : The January 2024 peak is not shown in the figure

- In Line 227 : You give results for males and females, but it would be a good idea to start by giving the characteristics of the participants in the methodology.

- In Line 244 : You forgot the parentheses on AH3N2

- In Figure 3 : Can you root the trees a, b and c as you rooted the others?

Reviewer 2 Report

Comments and Suggestions for Authors

The authors have performed reasonable study on epidemiological analyses and characterization of genomes of respiratory viruses co-infected with SARS-CoV-2. The article is well supported by data and justifications. The finding gave an insight into underlying mechanism to cause mutation in the mixed infections and helpful for the future studies. Here are a few suggestions:  

1. In order to understand the significance of this study, it would be appreciated if the authors could add in the introduction not only the background on the SARS-CoV-2 variants (lines 51-68), but also any previous studies that have elucidated the effects of SARS-CoV-2 on other respiratory viruses. Is the reference 16 the one at least?

2. Because the country information, Bulgaria, first appeared in the legend of Figure 1, please describe it first in the introduction, and then abstract and methodology. Additionally, if the author collected samples in different parts of the country, it would be helpful to indicate on a map where the samples were collected.

3. For Figure 1, please add information on the X and Y axes. Additionally, if the cumulative bar chart also shows the percentage of co-infections (%), it would be easier to read if the fluorescent green % co-infections were removed.

4. For Figure 2, please add information on the Y axis. Additionally, it would be better to show box-and-hide diagram instead of mean ± 1 SD.

5. line 615-616, please collect the way the references are written as [80,81]. 

Reviewer 3 Report

Comments and Suggestions for Authors

Here the authors analyzed the co-infection from SARS2 positive samples. They detected Influenza A/B virus and other 8 respiratory related viruses and analyzed the viral sequences. This is a well-designed study providing clues in clinical characteristics of respiratory virus infection. I thought the data was basically rigorous and several questions remain.

1.     Figure 2, the difference of CT values may be due to the PCR amplification efficiency of primers. The authors need provide evidence to show the identical amplification efficiency of primers with the same copy of templates. Or the authors can compare the same virus from different conditions.

2.     Can’t find qPCR information against Influenza virus. Besides, the authors need provide sequence information about primers and probes in tables.

3.     Line 36, Add coronaviruses, such as 229E, OC43.

4.     Line 142, ‘SARS-Vo-2’.

Round 2

Reviewer 3 Report

Comments and Suggestions for Authors

All concerns have been addressed.